# Native-to-invasive rodent species turn-over within African cities: The example of Niamey, Niger

Abdoulaziz Ibrahim Danzabarma[1,2]☉*, Karmadine Hima[1]☉, Madougou Garba[3], Seydou Issaka[1], Gauthier Dobigny[2,4]☉

1 Département de Biologie, Université Abdou Moumouni, Niamey, Niger, 2 Centre de Biologie pour la Gestion des Populations (UMR IRD-INRAE-CIRAD-Institut Agro-Université de Montpellier), Montferrier-sur-Lez, France, 3 Direction Générale de la Protection des Végétaux, Niamey, Niger, 4 Institut Pasteur de Madagascar, Unité Peste, Antananarivo, Madagascar

☉ These authors contributed equally to this work.
* az.danzabarma@gmail.com

## Abstract

Expanding transportation infrastructure has facilitated the introduction and proliferation of invasive small mammals, particularly now cosmopolitan rodents like *Rattus spp.* and *Mus musculus*, within urban socio-ecosystems where they may severely impact local biodiversity, economy and public health. However, fine-scale dynamics of such biological invasions are still poorly documented, especially in African cities where such biological invasions are still ongoing. We took advantage of a long-term monitoring program of terrestrial small mammal communities of Niamey, the capital city of Niger, to compare rodent and shrew faunas at the same sampling points ten years apart, namely in 2009–2013 and in 2020–2023. We show that *Rattus rattus* has been expanding significantly in almost all trapping areas across the city, while *Mastomys natalensis* relative abundance has decreased. This clear trend of native-to-invasive rodent species turnover was particularly marked within the core city, but some notable exceptions exist in peripheral as well as traditionally built zones where the native species still remains dominant. These patterns are discussed, with special attention given to possible human socio-economic and health implications.

## Introduction

Rodents represent the most diverse group of mammals, exhibiting a worldwide distribution across a wide range of environments on all continents [1,2]. Some species are highly adaptable, a trait that has enabled them to colonize a diverse panel of terrestrial habitats, even those that have been deeply human-modified [3,4]. In particular, a few cosmopolitan invasive species, such as *Rattus norvegicus, Rattus rattus* and *Mus musculus*, have been dominating small mammal communities in many urban areas worldwide [5,6] where they may have highly deleterious impacts on native

**Data availability statement:** All relevant data are within the paper and its Supporting Information files.

**Funding:** GD, SCARIA Project funded by Belmont Forum AID, Phd Thesis scholarship funded by IRD. The funders had no role in study design, data collection and analysis, decision to publish, or preparation of the manuscript.

**Competing interests:** The authors have declared that no competing interests exist.

species assemblages [7,8] as well as human societies through socio-economic burden [9] and public health issues [10,11].

However, Africa may represent an exception where rats and mice are currently still in the process of inland dissemination across human settlement networks, often through road and/or fluvial transports [12–19]. The displacement of the highly prolific and native rodent species, especially *Mastomys spp.*, by invasive black rats or house mice have been well documented in Senegal [14,15,17,19]. It is also apparent on the basis of ancient vs. more recent studies of rodents in some African cities. For instance, monitoring of small mammals in Dakar, Senegal, in the late 1920s [20] and 2016 [21] unambiguously show that native species were almost completely replaced by invasive ones, mostly house mice (*Mus musculus)*. Actually, available results from large city-scale trapping campaigns in Africa suggest that 99.0%, 96.8% and 65.0% of small terrestrial mammals (i.e., rodents and shrews) from Durban (South Africa), Makurdi (Nigeria) and Cotonou (Benin) belong to invasive species, namely house mice, black and/or Norway rats [22–24].

Niger is a semi-arid to arid landlocked West African country. The capital city, Niamey, lies on both sides of the Niger River. Major tarred roads link the city to Benin southward, Mali northward, Burkina-Faso westward, and to the rest of the country as well as Nigeria eastward. Extensive studies conducted between 2009 and 2013 on small mammals all over the city have shown that the rodent community was largely dominated by the native species *M. natalensis* which was widespread and abundant in most households [16]. However, the invasive species *Rattus rattus* and *Mus musculus* were found in a few restricted areas of the core city, essentially corresponding to industrial sites (black rats), markets and their surroundings (both species) where they were hypothesized to displace local rodents [16]. In addition, investigations based on large-scale rodent census, interviews and population genetics data [8,13] suggested that black rats and mice had been rather recently introduced, that invasion processes may still be ongoing, resulting from long-distance transports potentially from multiple geographic origins. If true, one expects that rats and mice may constitute important game changers, inducing drastic changes in small mammal community structure within Niamey with potential socio-economic and health implications.

In order to further document the dynamics and effects of this apparently ongoing rodent invasion process, we organized new small mammal monitoring campaigns between 2020 and 2023, notably targeting urban zones that had been sampled ten years before, and we compared them with the 2009–2013 results [16]. Doing so, we show a clear trend of black rats' spatial expansion in many areas of Niamey and we confirm that the species indeed tends to displace *M. natalensis*. As such, our study captures an unambiguous native-to-invasive rodent species turnover in an African city within a ten-year long period.

## Materials and methods

### Data available from 2009–2013 rodent trapping campaigns

In the 2009–2013 study (hereafter designed as to period P1; [16]), the term "localities" refers to neighborhoods of Niamey. Within each locality, the other's investigated

specific places referred to as "sites" where trapping was carried out. These sites are households, industrial complexes and/or market gardens. Here we follow their terminology in order to facilitate comparisons between the two studies. In their study, standard trapping sessions were conducted at 26 localities out of total 52 localities over the 2009–2013 period. These 26 localities included 18 neighborhoods, 6 market gardens and 2 industrial areas. Between 4–18 sites were sampled in each locality.

## New rodent trapping campaigns

Small mammal assemblages within human settlements have been extensively studied using morphological, cytogenetic, DNA sequencing and genotyping approaches in Niger in general, and in Niamey in particular (e.g., [13,25]), including those from the P1 campaigns [16]. As a consequence, diversity within the city is now very well described, allowing us to rely confidently on morphological characteristics as well as external measurements (weight, body and tail length, ear length and hindfoot length) as described by Granjon and Duplantier [26] to identify the specimens collected during the sampling campaign newly presented here.

During the 2020–2023 period (hereafter designed as to period P2), 21 localities were sampled (Table 1), including 13 ones that had already been investigated during P1 (see localities in bold in Table 1). For the latter neighborhoods monitored during both P1 and P2 campaigns, each time this was feasible, the same sites were re-sampled.

**Table 1. Localities sampled during the 2020-2023 (P2) campaign.**

| Acronym | Locality | Type | Effort (trapping night) | | | GPS | |
|---|---|---|---|---|---|---|---|
| | | | Wm | Sh | Total | Lat. (N) | Lon. (E) |
| AER | AEROPORT | RA | 112 | 104 | 216 | 13.51222 | 2.12138 |
| BOB | BOBIEL | RA | NA | NA | NA | 13.56076 | 2.09214 |
| **BOU** | **BOUKOKI** | **RA** | **595** | **162** | **757** | **13.53868** | **2.11395** |
| **DAR** | **DAR ES SALAM** | **RA** | **827** | **89** | **916** | **13.54693** | **2.09528** |
| **GAM** | **GAMKALLEY** | **RA** | **426** | **426** | **852** | **13.49388** | **2.12693** |
| **KAR** | **KARADJE** | **RA** | **78** | **36** | **114** | **13.49387** | **2.09441** |
| KIR | KIRKISSOYE | F | 28 | 17 | 45 | 13.49500 | 2.10961 |
| KIR-1 | KIRKISSOYE 1 | RA | 6 | 3 | 9 | 13.48109 | 2.11656 |
| **KOT** | **KOIRA TEGUI** | **RA** | **30** | **40** | **70** | **13.58225** | **2.11442** |
| **CYA** | **CORNICHE YANTALA** | **RA** | **267** | **96** | **363** | **13.51262** | **2.09911** |
| LAC | LACOUROUSSOU | RA | 62 | 82 | 144 | 13.51222 | 2.12138 |
| LMO | LAMORDE | RA | NA | NA | NA | 13.50605 | 2.07266 |
| **GRM** | **LIBERTE** | **RA** | **97** | **62** | **159** | **13.51923** | **2.11518** |
| NYA | NIAMEY 2000 | RA | NA | NA | NA | 13.51427 | 2.18560 |
| PLT | PLATEAU | RA | NA | NA | NA | 13.54083 | 2.10273 |
| **TCH** | **TCHANGAREY** | **RA** | **30** | **102** | **132** | **13.58029** | **2.08317** |
| UAM | UAM | RA | NA | NA | NA | 13.49918 | 2.10535 |
| **WAD** | **WADATA** | **RA** | **35** | **32** | **67** | **13.51821** | **2.14419** |
| SNI | SONUCI | RA | 260 | 90 | 350 | 13.56429 | 2.07065 |
| LSG | LOSSA SOUNGOU | RA | NA | NA | NA | 13.54395 | 2.04551 |
| YAH | YANTALA-HAUT | RA | NA | NA | NA | 13.53435 | 2.08208 |

For each locality, the type of environment (RA = residential area, F = factory) as well as the trapping effort (i.e., the number of night-traps) using only Sherman (Sh), only locally wire mesh (Wm) and both types of traps (Total) are indicated. Localities that could be statistically compared between the 2009–2013 (P1) and 2020–2023 (P2) campaigns are indicated in bold (see text for details). The term NA is used for localities where the trapping effort is not known with certainty.

Additional data from extra localities were collected across Niamey during the P2 period, thus allowing us to document further the extant distributions of native and invasive species observed to date in Niamey. That may be of interest for other purposes, including eco-evolutionary meta-analyses, niche modeling, zoonotic pathogen screenings, etc.

In each investigated space (courtyards, grain mills, dwellings and/or business premises such as stores or workshops), from two to four traps (depending on the space available within one given space) were set during three consecutive nights. They were set in the afternoon, and checked in the morning: any successful trap was replaced by a new baited trap laid at the exact same place, while unsuccessful traps were rebaited and reset for another night. Two models of traps, namely Sherman© (here after designed as to "Sh"; 8x9x23 cm; H.B. Sherman Traps, Inc., Tallahassee, FL, USA) and locally manufactured wire mesh ("Wm"; 10x10x25 cm) traps were used since they were shown to show significantly different success rate depending on the species, with Wm being more successful at capturing *Rattus spp.* while Sh were more performant for *M. natalensis*, *Mus musculus* and the native shrew *Crocidura olivieri* [16,27,28]. The baits were made of peanut butter and smoked fish or sardines usually combined with "soumbala," a local cooking powder made of *Parkia biglobosa* seeds. Traps were set inside one factory (rice industry in Kirkissoye, KIR) and residential structures within 20 different localities.

Rodents were captured and brought alive to the laboratory in their traps immediately after capture. They were sacrificed within the next 3–4 hours following cervical dislocation in accordance with the guidelines of the American Society of Mammalogists about rodent handling, process and euthanasia (Mills et al., 1995; Sikes et al., 2011). Observations and measurements (weight, body measurements, sex, reproductive activity, possible ecto-parasites) were systematically carried out afterwards. Tissue samples were also collected for future molecular analyses (species-specific barcoding identification, pathogens detection, population genetics, etc.). Full details regarding the localities investigated as well as the trapping effort for each type of traps are provided in Table 1.

### Ten-year evolution of rodent communities

To assess long-term changes in urban rodent communities over a ten-year period, we compared trapping data collected during two distinct campaigns: P1 (2009–2013) and P2 (2020–2023). The P1 campaign spanned two years, while P2 extended over three years. Given the logistical constraints associated with large-scale urban sampling (i.e., 987 individuals from 52 intra-Niamey neighborhoods during P1 [16], and 604 captures from 21 neighborhoods during P2 [this study]), it was not feasible to sample all localities within the same months or seasons during the same campaigns. As a result, trapping sessions occurred at different times of the year. However, due to the absence of significant seasonal migration patterns in urban rodent populations, data collected across different seasons within each campaign could be pooled.

To facilitate a meaningful comparison of small mammal assemblages between P1 and P2 trapping periods, only sites that had been accurately monitored in terms of trapping effort and types of traps during both P1 and P2 campaigns were considered. This enabled a robust analysis of trap- and species-specific trapping success in nine given localities, namely BOU, CYA, DAR, GAM, GRM, KAR, KOT, TCH and WAD (Fig 1).

### Statistical analyses

To account for variations in trapping effort across localities and periods, species- and trap-specific capture rates were calculated. These rates were determined by dividing the number of individuals captured by the number of trap-nights for each trap type.

Species-specific trapping successes retrieved during the P2 campaign were compared between Wm and Sh traps over all localities using t-tests. Since Wm and Sh traps were confirmed to display significantly different species-specific capture rates (see Results), both species and trap type were taken into account in all subsequent statistical comparisons.

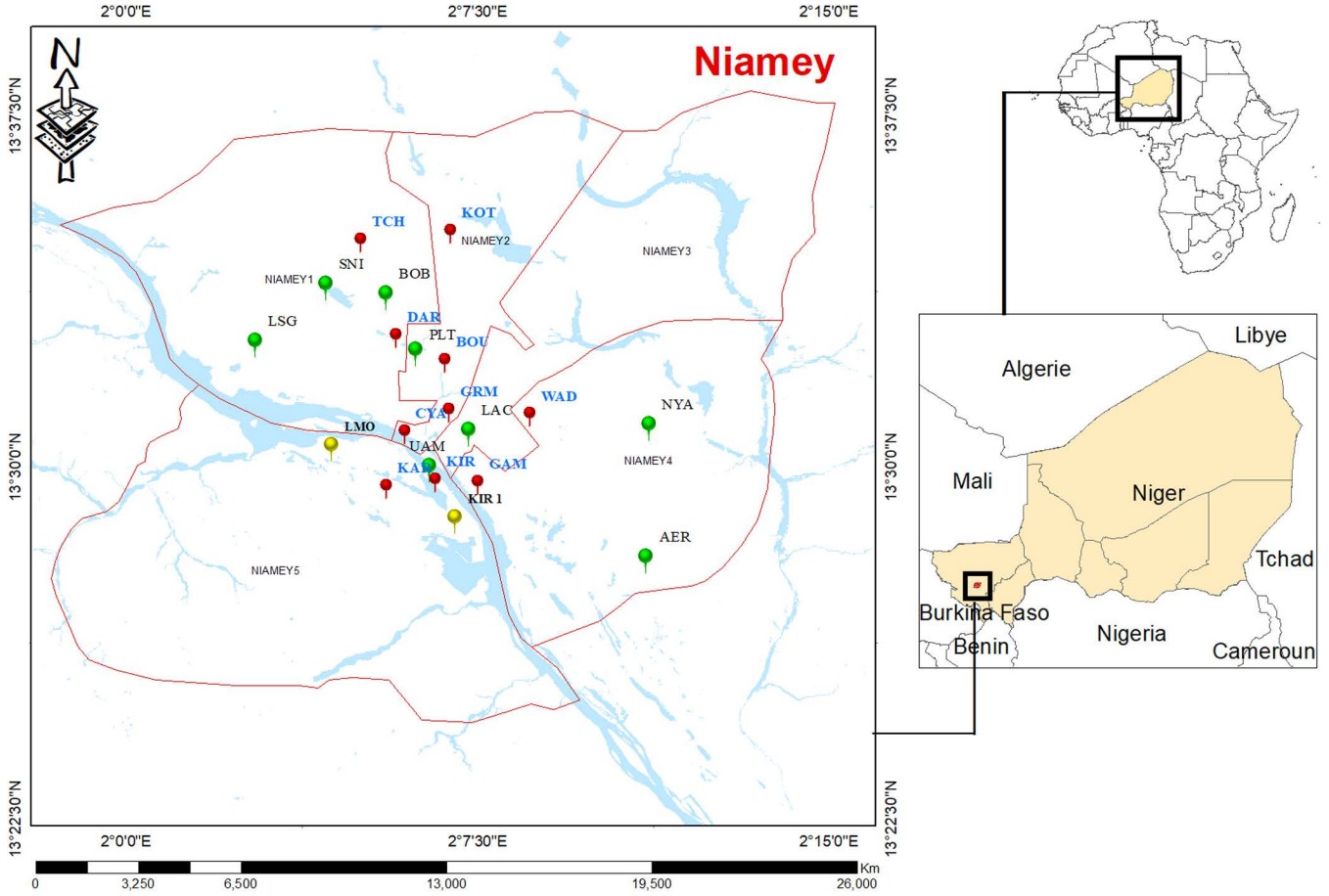

**Fig 1. Study area map of Niamey, Niger, showing sampled localities across two periods.** The red lines denote the boundaries of the five municipal districts. Watercourses are shown in blue The green dots indicate localities sampled only in P2. Localities sampled in both P1 and P2 are represented on red if sampled sites are the same between the two periods and on yellow if the sites are different.

Trap- and species-specific capture rates were compared over all available localities between P1 and P2 campaigns using pairwise Holm-adjusted Wilcoxon tests. When feasible (i.e., for localities where at least one individual was captured), trap- and species-specific capture rates were also compared at the locality level using the same procedure.

## Ethical and data sharing

This study was part of a collaboration between Abdou Moumouni University of Niamey (UAM) and French Institute of Research for Sustainable Development (IRD) under the scientific partnership agreement number 301027/00. Explicit oral consent was obtained from each locality authorities as well as each household head where trapping was implemented. None of the small mammal species sampled here has protection status [29]. All captured animals were treated humanely, taking into account their individual welfare, in accordance with the guidelines of the American Society of Mammalogists [30] (Sikes, Care, and Mammalogists 2016). Ethical clearance was obtained from the Ministry for Public Health of the Republic of Niger (permit N°054/2022/CNERS). Prior to rodent trapping per se, the study's objectives, methodology, and potential implications were thoroughly explained to neighborhood chiefs, who appointed local representatives to facilitate communication with household heads. Verbal informed consent was explicitly obtained from each household head before any trapping activities took place, ensuring full compliance with ethical research standards and community collaboration.

## Results

### Assessment of small mammal diversity and trap type effect

A total of 604 small mammals belonging to three rodent (*Rattus rattus, Mastomys natalensis* and *Mus musculus*) and one shrew (*Crocidura olivieri*) species were collected during the P2 campaign (Table 2). Black rats were the most abundant (N = 315), followed by *M. natalensis* (N = 213), house mice (N = 46) and shrews (N = 30).

Comparison of capture rates during the P2 period unambiguously confirmed that Wm and Sh traps displayed significantly different results depending on the species considered. As such, Wm was found more effective at capturing *R. rattus* (Wm = 263 captures/ 4368 night-traps vs. Sh = 29/2943; t = 2.109, p = 0.036), while Sherman traps were more suitable for capturing *M. musculus* (Wm = 5/245 vs. Sh = 91/221; t = −5.0341, p < 0.001), *M. natalensis* (Wm = 92/4368 vs. Sh = 318/2945; t = −5.272, p < 0.001) and *C. olivieri* (Wm = 36/4368 vs. Sh = 288/2943; t = −2.1753, p = 0.031). These differences in species-specific capture rates depending of the trap type justified to conduct subsequent analyses by considering Wm and Sh traps separately.

### Evolution of species-specific spatial distributions between P1 and P2 periods

Among the nine localities investigated during both the P1 and P2 campaigns (Table 1), *Rattus rattus* was captured in only two of them (CYA and GRM) during P1 (Fig 2A; Table 3), with a single and three captures, respectively, despite high trapping efforts [16]. During the P2 period, the species was captured in seven of these same nine locations, namely BOU,

**Table 2. Trapping effort and number of captures per locality for the 2020-2023 period (P2).**

| Locality | Effort (trapping night) | | | Number of captures | | | | |
|---|---|---|---|---|---|---|---|---|
| | Wm | Sh | total | *Cro* | *Mna* | *Mmu* | *Rra* | Total |
| AER | 112 | 104 | 216 | 2 | 21 | 0 | 0 | 23 |
| BOU | 595 | 162 | 757 | 6 | 17 | 0 | 118 | 141 |
| CYA | 30 | 40 | 70 | 1 | 14 | 0 | 21 | 36 |
| DAR | 267 | 96 | 363 | 15 | 1 | 0 | 80 | 96 |
| GAM | 827 | 89 | 916 | 0 | 80 | 0 | 2 | 82 |
| GRM | 97 | 62 | 159 | 0 | 0 | 35 | 26 | 61 |
| KAR | 426 | 426 | 852 | 0 | 3 | 0 | 20 | 23 |
| KIR | 78 | 36 | 114 | 0 | 0 | 0 | 8 | 8 |
| KIR-1 | 28 | 17 | 45 | 0 | 0 | 0 | 4 | 4 |
| KOT | 6 | 3 | 9 | 0 | 16 | 0 | 0 | 16 |
| LAC | 62 | 82 | 144 | 0 | 0 | 11 | 0 | 11 |
| SNI | 260 | 90 | 350 | 2 | 2 | 0 | 0 | 4 |
| TCH | 30 | 102 | 132 | 4 | 20 | 0 | 0 | 24 |
| WAD | 35 | 32 | 67 | 0 | 1 | 0 | 14 | 15 |
| BOB | NA | NA | NA | 0 | 0 | 0 | 5 | 5 |
| LAM | NA | NA | NA | 0 | 6 | 0 | 0 | 6 |
| LSG | NA | NA | NA | 0 | 25 | 0 | 0 | 25 |
| NYA | NA | NA | NA | 0 | 6 | 0 | 0 | 6 |
| PLT | NA | NA | NA | 0 | 0 | 0 | 5 | 5 |
| UAM | NA | NA | NA | 0 | 0 | 0 | 5 | 5 |
| YAH | NA | NA | NA | 0 | 1 | 0 | 7 | 8 |
| | 2853 | 1341 | 4194 | 30 | 213 | 46 | 315 | 604 |

"Wm" and "Sh" correspond to wire-mesh and Sherman traps, respectively. "Cro", "Mna", "Mmu" and "Rra" stand for *Crocidura olivieri*, *Mastomys natalensis*, *Mus musculus*, and *Rattus rattus*, respectively.

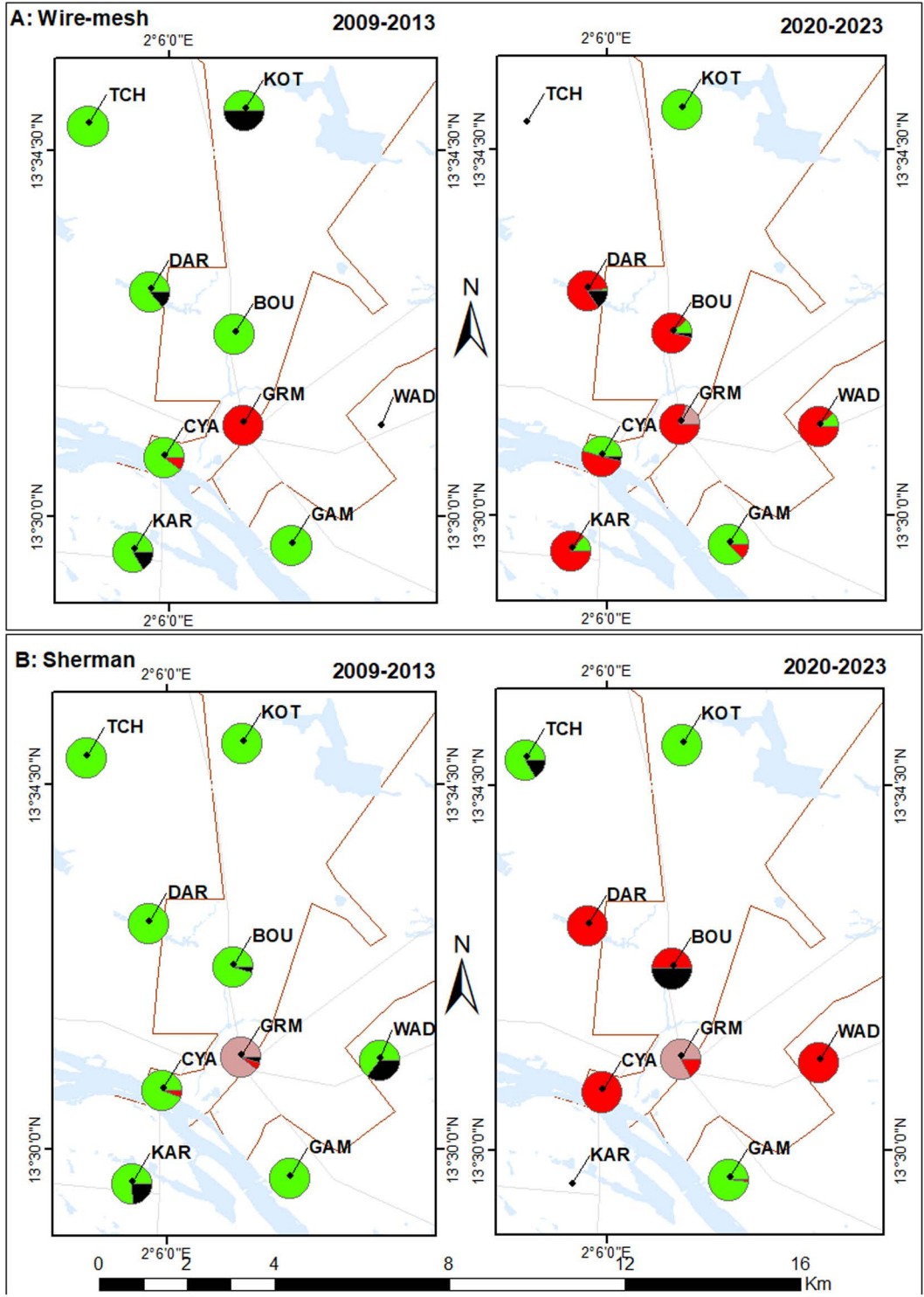

**Fig 2. Relative species abundances during periods P1 and P2 by trap type.** This figure presents the relative abundances of small mammal species recorded during sampling periods P1 (left panel) and P2 (right panel), according to trap type: wire mesh traps (A) and Sherman traps (B). Data are displayed as pie charts, with each species represented by a specific color: green for *Mastomys natalensis*, brown for *Mus musculus*, red for *Rattus rattus*, and black for *Crocidura olivieri*.

**Table 3. Comparisons of trapping efforts, capture number and rates for each species between the P1 and P2 campaigns.**

| Species | Locality | Traps model | | | | | | | | | | | | Overall |
|---|---|---|---|---|---|---|---|---|---|---|---|---|---|---|
| | | Wire-mesh traps | | | | | | Sherman traps | | | | | | |
| | | 2009-2013 | | | 2020-2023 | | | 2009-2013 | | | 2020-2023 | | | |
| | | TN | N | CR | TN | N | CR | TN | N | CR | TN | N | CR | |
| Rattus rattus | BOU | 220 | 0 | 0 | 595 | 117 | 0.197 | 229 | 0 | 0 | 162 | 1 | 0.006 | 292 |
| | CYA | 247 | 1 | 0.004 | 267 | 16 | 0.06 | 241 | 3 | 0.012 | 96 | 5 | 0.052 | |
| | DAR | 279 | 0 | 0 | 827 | 78 | 0.094 | 252 | 0 | 0 | 89 | 2 | 0.022 | |
| | GAM | 236 | 0 | 0 | 426 | 1 | 0.002 | 224 | 0 | 0 | 426 | 1 | 0.002 | |
| | GRM | 148 | 3 | 0.02 | 97 | 20 | 0.206 | 157 | 4 | 0.025 | 62 | 6 | 0.097 | |
| | KOT | 135 | 0 | 0 | 30 | 0 | 0 | 131 | 0 | 0 | 40 | 0 | 0 | |
| | TCH | 134 | 0 | 0 | 30 | 0 | 0 | 96 | 0 | 0 | 102 | 0 | 0 | |
| | KAR | 337 | 0 | 0 | 78 | 20 | 0.256 | 318 | 0 | 0 | 36 | 0 | 0 | |
| | WAD | 247 | 0 | 0 | 35 | 7 | 0.2 | 250 | 0 | 0 | 32 | 7 | 0.219 | |
| | subtotal | 1714 | 4 | 0.002 | 2325 | 259 | 0.111 | 1671 | 7 | 0.004 | 903 | 22 | 0.024 | |
| Mastomys natalensis | BOU | 220 | 7 | 0.032 | 595 | 17 | 0.029 | 229 | 41 | 0.179 | 162 | 0 | 0 | 410 |
| | CYA | 247 | 9 | 0.036 | 267 | 14 | 0.052 | 241 | 52 | 0.216 | 96 | 0 | 0 | |
| | DAR | 279 | 6 | 0.022 | 827 | 1 | 0.001 | 252 | 34 | 0.135 | 89 | 0 | 0 | |
| | GAM | 236 | 3 | 0.02 | 426 | 20 | 0.206 | 224 | 4 | 0.025 | 426 | 73 | 0.097 | |
| | GRM | 148 | 0 | 0 | 97 | 0 | 0 | 157 | 0 | 0 | 62 | 0 | 0 | |
| | KAR | 337 | 10 | 0.03 | 78 | 3 | 0.038 | 318 | 39 | 0.123 | 36 | 0 | 0 | |
| | KOT | 135 | 2 | 0.015 | 30 | 7 | 0.233 | 131 | 8 | 0.061 | 40 | 9 | 0.225 | |
| | TCH | 134 | 5 | 0.037 | 30 | 0 | 0 | 96 | 11 | 0.115 | 102 | 20 | 0.196 | |
| | WAD | 247 | 0 | 0 | 35 | 1 | 0.029 | 250 | 11 | 0.044 | 32 | 0 | 0 | |
| | Subtotal | 1835 | 42 | 0.023 | 2288 | 50 | 0.022 | 1741 | 216 | 0.124 | 983 | 102 | 0.104 | |

| Species | Locality | Traps model | | | | | | | | | | | | Overall |
|---|---|---|---|---|---|---|---|---|---|---|---|---|---|---|
| | | Wire-mesh traps | | | | | | Sherman traps | | | | | | |
| | | 2009-2013 | | | 2020-2023 | | | 2009-2013 | | | 2020-2023 | | | |
| | | TN | N | CR | TN | N | CR | TN | N | CR | TN | N | CR | |
| Crocidura olivieri | BOU | 220 | 0 | 0 | 595 | 5 | 0.008 | 229 | 2 | 0.009 | 162 | 1 | 0 | 54 |
| | CYA | 247 | 0 | 0 | 267 | 1 | 0.004 | 241 | 0 | 0 | 96 | 0 | 0 | |
| | DAR | 279 | 1 | 0.004 | 827 | 15 | 0.018 | 252 | 0 | 0 | 89 | 0 | 0 | |
| | GRM | 148 | 0 | 0 | 97 | 0 | 0 | 157 | 3 | 0.019 | 62 | 0 | 0 | |
| | KAR | 337 | 2 | 0.006 | 78 | 0 | 0 | 318 | 12 | 0.038 | 36 | 0 | 0 | |
| | KOT | 135 | 2 | 0.015 | 30 | 0 | 0 | 131 | 0 | 0 | 40 | 0 | 0 | |
| | TCH | 134 | 0 | 0 | 30 | 0 | 0 | 96 | 0 | 0 | 102 | 4 | 0.039 | |
| | WAD | 247 | 0 | 0 | 35 | 0 | 0 | 250 | 6 | 0.024 | 32 | 0 | 0 | |
| | subtotal | 1747 | 5 | 0.003 | 1959 | 21 | 0.011 | 1674 | 23 | 0.014 | 619 | 5 | 0.008 | |
| Mus musculus | BOU | 220 | 0 | 0 | 595 | 0 | 0 | 229 | 0 | 0 | 162 | 0 | 0 | 96 |
| | CYA | 247 | 0 | 0 | 267 | 0 | 0 | 241 | 0 | 0 | 96 | 0 | 0 | |
| | DAR | 279 | 0 | 0 | 827 | 0 | 0 | 252 | 0 | 0 | 89 | 0 | 0 | |
| | GRM | 148 | 0 | 0 | 97 | 5 | 0.052 | 157 | 61 | 0.389 | 62 | 30 | 0.484 | |
| | KAR | 337 | 0 | 0 | 78 | 0 | 0 | 318 | 0 | 0 | 36 | 0 | 0 | |
| | KOT | 135 | 0 | 0 | 30 | 0 | 0 | 131 | 0 | 0 | 40 | 0 | 0 | |
| | TCH | 134 | 0 | 0 | 30 | 0 | 0 | 96 | 0 | 0 | 102 | 0 | 0 | |
| | WAD | 247 | 0 | 0 | 35 | 0 | 0 | 250 | 0 | 0 | 32 | 0 | 0 | |
| | subtotal | 148 | 0 | 0 | 97 | 5 | 0.052 | 157 | 61 | 0.389 | 62 | 30 | 0.484 | |
| Overall | | | 51 | | | 335 | | | 307 | | | 159 | | 852 |

These data are organized according to trap types and localities."TN" stand for the number of trap-night (i.e., the total number of nights during which traps were set, representing the trapping effort), "N" refers to the number of captures, and "CR" represents the capture rate (i.e., the ratio of captures to trap-nights, used as a measure of trapping success or yield).

CYA, DAR, KAR, GAM, GRM and WAD (Fig 2A; Table 3). When one considers captures obtained with both Sh and Wm traps, *M. natalensis* was found in eight localities during both periods P1 and P2 (Fig 2B; Table 3). Despite the extensive trapping effort throughout the city ([16]; this study), *M. musculus* was captured only in GRM. However, Sh-based results unambiguously demonstrate that the species was abundant in this given locality during both P1 and P2 campaigns (Fig 2B; Table 3). Finally, the shrew *C. olivieri* was present in the P1 but absent from the P2 datasets from several localities, namely, including three (KAR, GRM, and WAD).

In brief, from the spatial distribution perspective, *R. rattus* has been found in two then seven localities during the P1 and P2 campaigns, respectively, while *M. natalensis* was present in all nine localities during both periods. House mice were present but restricted to one single locality during the two campaigns. Shrews were found in six and four localities during P1 and P2, respectively.

## Comparison of species- and trap-specific capture rates between P1 and P2 campaigns

Taking trap type into consideration, we explored the differences in species-specific capture rates between P1 (2009–2013) and P2 (2020–2023) campaigns (see detailed results in Table 3 and S1 Table). At the city scale, *Rattus rattus* capture rates exhibited a pronounced and significant increase for both wire mesh (0.2% during P1, to 11% during P2; $p < 0.001$) and Sherman traps (0.4% to 2.4%; $p = 0.049$). Upon examination of the locality level, the increase observed with Wm traps was significant for all (BOU, CYA, CAR, GRM and KAR; all $p < 0.037$, S1 Table) but two localities (GAM and WAD, the latter showing only marginally significant increase, $p = 0.053$, S1 Table). Systematic but non-significant increases were also observed for all localities with Sh-based data.

Conversely, a slight but highly significant decrease of *M. natalensis* capture rate was observed at the city scale for Sh traps (12.4% to 10.4%; $p < 0.001$) between P1 and P2 periods. This decrease was lower and non-significant with Wm traps (2.3% to 2.2%; $p = 0.22$). When looking at the locality level, no significant difference was observed with Wm between P1 and P2 for any localities ($0.13 < p < 0.95$) except one (KOT, $p = 0.01$). However, highly significant differences were retrieved with Sh (all $p < 0.007$), except for GAM ($p = 0.18$), KOT ($p = 0.1$) and TCH ($p = 0.79$) which showed non-significant decreases (S1 Table).

Although the capture rates of the indigenous species *C. olivieri* was always rather low, it was significantly greater during P1 than P2 when only Sh traps were considered at the city level ($p = 0.0046$). However, no difference was observed between the two periods for Wm traps ($p = 0.085$). At the locality level, these differences were less clear since there was a significant decrease for Sh only in KAR (from 3.8% to 0%; $p = 0.035$) but no significant changes in the other localities (BOU, GRM, TCH and WAD, all $p < 0.1$). A very slight increase was found significant with Wm traps in BOU (0% to 0.008%; $p = 0.045$) while no change was observed in the other localities (CYA, DAR, KAR and KOT; all $p > 0.11$).

House mice were captured during both P1 and P2 only in GRM, and no difference were observed in capture rates in this locality, whatever the type of traps used (both $p > 0.14$).

## Discussion

The originality of our study lies in the fact that the exact same trapping localities and sometimes sites were monitored at an interval of ten years in order to investigate possible changes in the compositional changes of commensal rodent communities within an African city, i.e., Niamey, Niger, in the context of the hypothesized gradual spread of two invasive species, namely the black rat *R. rattus* and the house mouse *M. musculus*.

As already highlighted in other studies (e.g., [16,27,28]), our data unambiguously confirm that different types of traps may display significantly different capture rates, thus once again showing the importance of trapping protocols for small mammal biodiversity assessment and analyses. This appeared particularly critical here when addressing the interrelationships between Wm-attracted rats vs. Sh-attracted *M. natalensis* and *M. musculus*.

We also show that the house mouse remains present in the only locality where it was already present ten years ago, but that it did not appear to expand massively throughout the city since then. During the 2020–2023 campaigns, this prolific species was also captured in LAC (Table 2), a neighborhood located 1 km apart from the first one (see Fig 2), but in absence of LAC in the 2009–2013 dataset [16], we cannot argue about any spatial expansion. To our knowledge, there is no available data about the date of *M. musculus* introduction into Niamey. As a consequence, it is not known how long the species has been present in the city. Nevertheless, taking into account the invasion success of house mice in other major Sahelian cities (e.g., Dakar, Senegal: [21]; Bamako, Mali: Granjon et al., in press), we anticipate that this species will spread further within Niamey in the years and decades to come.

Conversely, our longitudinal survey unambiguously demonstrates that black rats have already expanded widely across the city, with its presence now ascertained in at least five new neighborhoods (BOB, PLT, UAM, YAH et WAD). Our data thus clearly support Garba and colleagues' [16] hypothesis that *R. rattus* active bio-invasion is currently ongoing in Niamey. The species have spread for several kilometers across the urban landscape during the last ten years, thus showing a successful colonization of the city. Interestingly, it has progressed mainly within the core city, while we did not observe it yet in the more peripheral and more recently urbanized areas, like KOT and TCH. It is also noticeable that, although black rats were identified in Gamkalleye (GAM) during the P2 campaign while they were not found there during the P1 one, this neighborhood may partly resist black rats' invasion. Indeed, additional regular surveys conducted within this particular area in 2023 and 2024 tend to suggest that *R. rattus* does not increase neither in abundance nor in terms of infested households (our own unpublished data). Gamkalleye is one of the last traditionally built districts of Niamey, with a large majority of houses still made of mud (i.e., the so-called "banco" material). Though densely built and populated, and located in the heart of Niamey, this now atypical urban area may keep rural-like features (e.g., sandy soils, mud-made walls, frequent presence of livestock breeding) that could at least partly favor the native *M. natalensis* over the recently introduced *R. rattus* species. This particular situation in Gamkalleye is still under longitudinal surveillance. However, the rapid dissemination of black rats in the core city on the one hand, and its absence or rareness in both peripheral (KOT, TCH) and traditionally built (GAM) zones on the other hand, strongly suggest that, in Niamey, black rat expansion primarily follows formal, usually hard-built, urbanization. However, the eco-evolutionary processes (e.g., competition for resources, parasite sharing, environment-mediated reproductive rate) and/or socio-ecosystem characteristics (e.g., human action against pest, human food provisioning) that explain such different dynamics of *Rattus rattus*– *Mastomys natalensis* remain to be investigated [38].

In addition to its expanding spatial distribution, our quantitative comparisons of trap-specific capture rates at the neighborhood levels suggest that *R. rattus* has also locally increased in relative abundance. In parallel, we found that *M. natalensis* tends to decrease in several settings where black rat was present. These trends have been statistically tested with two different methods whose results were both congruent (r = −0.67; S1 Fig). Together, these patterns strongly support the replacement of the former by latter species within Niamey, as already hypothesized by Garba and colleagues [16]. This native-to-invasive rodent displacement is further exemplified by the apparently achieved turnover observed in DAR where *M. natalensis* were abundant in 2009–2013 but are absent from the 2020–2023 dataset, while *R. rattus* was initially absent and are now dominant (Fig 2).

Of course, such drastic changes in terrestrial small mammal faunas within cities have already existed, as shown by the extant rodent assemblages of large European or American cities where only invasive, now cosmopolitan taxa (e.g., *Rattus rattus*, *R. norvegicus* and *M. musculus*) are present in hard-built environments. However, the process may be currently ongoing in Niamey, and this may raise several issues in a near future. In particular, proliferation of invasive animal reservoirs like rats may propagate new zoonotic pathogens or increase the prevalence of already circulating ones [31], especially within urban environments [11,32]. In several West African towns and villages, including in Niger, invasive black rats have already been suspected to fuel atypical trypanosomes dissemination [33,34]. Some authors have also pointed towards the risk of Seoul virus propagation in this part of the world following the spillover of this deadly virus from its usual reservoir *Rattus norvegicus* to its congeneric *R. rattus* [35]. Conversely, *M. natalensis* is known as the major reservoir

for Lassa virus [36], although some rare instances of alternative hosts -which do not include *Rattus rattus*- have been described [37]. As a consequence, shifts in reservoir species assemblages as that described in the present study are expected to have important sanitary consequences in Sahelian cities: while some zoonotic pathogens may become rarer (e.g., Lassa virus), others may appear and cause human diseases that are essentially unknown, hence undiagnosed and untreated, in this region (e.g., Orthohantavirus fevers, atypical trypanosomiais). A wide panel of other infectious threats could also be impacted (e.g., leptospirosis, bartonellosis, helminth- and bacteria-mediated diarrhea, antibio-resistance evolution, etc.) and may deserve further attention [32–37].

Beyond health, rodents are also responsible for various socio-economic nuisances. In Niamey, a study on the knowledge and perception of rodents and rodent-associated issues revealed that 96.5% of 170 interviewed inhabitants from 18 different neighborhoods identified rodents as pests [38]. All over the city, people quoted rodent-induced damages on food and food stocks (63.1%), houses infrastructures (47.3%) and furniture (19.5%) as well as clothes (16.8%), thus suggesting that rodents strongly reinforce economic vulnerability and represent a significant additional burden for inhabitants [38]. These data were collected from end of 2009 to May 2011, a period where *M. natalensis* was still over-dominant across most of Niamey [38]. However, black rats are also widely recognized as important pest rodents that may greatly damage crops, food stocks and household utensils [39]. As such, while rats progressively replace native *M. natalensis* within Niamey, we expect nuisances to persist but potentially to change in nature and/or intensity. Unfortunately, we are not aware of studies that have compared *M. natalensis* vs. *R. rattus* damages, or their perception by inhabitants, that would allow one to anticipate future socio-economic nuisances induced by the native-to-invasive rodent turnover documented here.

## Supporting information

**S1 Table. Results of statistical tests of capture rates comparisons between periods.** Data are organized by trap type and locality. The term "Stats" refers to the use of Holm-adjusted two-tailed Wilcoxon test p-values. Where possible (i.e., for localities where at least one individual was captured), capture rates by trap and species were also compared at locality level using the same procedure.
(DOCX)

**S1 Fig. Correlation between *R. rattus* and *M. natalensis* trapping effort and capture number.** The panel on the right shows the evolution of the ratio [(*R. rattus* – *M. natalensis*)/Nb of trap-nights] between the two periods. The red dots indicate the ratios of period 1 while the blues indicate those of period 2. the line at 0 separates the positives (above) from the negatives (below) values. The left-panel shows the Spearman correlation graph between the trapping success of the two species.
(TIF)

## Acknowledgments

The long-term monitoring of rodents within the city of Niamey, Niger, has been supported since 2013 by the West African network of small mammal observatories (ObsMICE) coordinated by the French Institute for Sustainable Research (IRD). This work was part of the SCARIA project (coord. G. Dobigny, Belmont Forum, "Pathways to Sustainability" call, 2021–2023) with the studies conducted in Niger funded by the South African National Research Fund. Abdoulaziz Danzabarma is supported by a PhD thesis bursary by the French Institute of Sustainable Development (IRD; convention signed between Mr Danzabarma, IRD and the Abdou Moumouni University of Niamey on the 22nd October 2022).

## Author contributions

**Conceptualization:** Abdoulaziz Ibrahim Danzabarma, Karmadine Hima, Gauthier Dobigny.

**Formal analysis:** Abdoulaziz Ibrahim Danzabarma, Gauthier Dobigny.

**Funding acquisition:** Gauthier Dobigny.

**Investigation:** Abdoulaziz Ibrahim Danzabarma, Karmadine Hima, Madougou Garba, Seydou Issaka, Gauthier Dobigny.

**Methodology:** Abdoulaziz Ibrahim Danzabarma, Karmadine Hima, Gauthier Dobigny.

**Supervision:** Karmadine Hima, Madougou Garba, Gauthier Dobigny.

**Writing – original draft:** Abdoulaziz Ibrahim Danzabarma, Karmadine Hima, Gauthier Dobigny.

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
