## [Decision Letter · Decision Letter 0]

Dear Dr. Ibrahim Danzabarma,

Thank you for submitting your manuscript to PLOS ONE. After careful consideration, we feel that it has merit but does not fully meet PLOS ONE’s publication criteria as it currently stands. Therefore, we invite you to submit a revised version of the manuscript that addresses the points raised during the review process.

We look forward to receiving your revised manuscript.

Kind regards,

Javier delBarco-Trillo, Ph.D.

Academic Editor

PLOS ONE

2. Please provide captions for Fig. 4 in your manuscript.

3. In the ethics statement in the Methods, you have specified that verbal consent was obtained. Please provide additional details regarding how this consent was documented and witnessed, and state whether this was approved by the IRB.

 [GD, SCARIA Project funded by Belmont Forum

AID, Phd Thesis scholarship funded by IRD]. 

6. We note that your Data Availability Statement is currently as follows: [All relevant data are within the manuscript and its Supporting Information files.]

Additional Editor Comments:

I have received comments from three reviewers. The combination of all of these comments call for a significant revision of the manuscript.

Reviewers' comments:

Reviewer's Responses to Questions

**Comments to the Author**

1. Is the manuscript technically sound, and do the data support the conclusions?

Reviewer #1: Yes

Reviewer #2: Yes

Reviewer #3: Partly

2. Has the statistical analysis been performed appropriately and rigorously?

Reviewer #1: Yes

Reviewer #2: Yes

Reviewer #3: No

3. Have the authors made all data underlying the findings in their manuscript fully available?

Reviewer #1: Yes

Reviewer #2: Yes

Reviewer #3: Yes

4. Is the manuscript presented in an intelligible fashion and written in standard English?

Reviewer #1: Yes

Reviewer #2: Yes

Reviewer #3: Yes

Reviewer #1: Dear authors, congratulations on your work! It was a pleasure to read it, very well explained and justified. I just miss a bit of attention in the graphs, I think they could be improved. Also, I suggested in the comments transforming a table into a barplot so we can see more clearly the differences in your results, since you just tested differences with t-tests. I think these would largely improve your work.

Reviewer #3: In ‘Native-to-invasive rodent species turn-over within African cities: the example of Niamey, Niger’, the authors compare small mammal presence and abundance at 9 localities in Niamey sampled about 10 years apart and conclude that R. rattus presence is expanding in the city, while M. natalensis is less abundant but equally prevalent. This is a simple study with sparsely described methods that makes a small contribution to our understanding of the expansion of an invasive rodent in urban areas over time; however, the statistical analyses are such that strong conclusions cannot be made. In addition, there seems to be a missed opportunity here to delve more deeply into the urban conditions specifically associated with Rattus expansion/abundance or to tease apart potential drivers of this expansion. Some detailed comments follow?

Line 38 – “rats and mice” are not species names and can encompass a wide range of animals with different life history traits. It would be preferrable to be specific here.

Line 42 – The provided references do not support the claim that the continent of Africa is any different from South America (for example) in terms of colonization of invasive rodents. This study doesn’t require that Africa be unique in this way for the paper to be of interest, so can be removed.

Line 49 – Please clarify what “essentially house mice” means.

Line 50 – Please use consistent numbers of significant digits for % throughout

Line 55 – Suggest referring to the map in Figure 1 here.

Lines 56-67 – Please indicate which (if any) of these studies were performed by authors of the current MS – especially Ref 16, which does seem to include some of the same names. It’s a bit confusing the way this part and the beginning part of the methods are written as to if this study followed the same methodology consistently between studies or just attempted to replicate the methodology as it appeared in the original publications.

Methods – Please provide details about what happened to the rodents during and after trapping. This is a significant oversight in methodology and should include (at a minimum) how long the traps were open for, how often they were checked, how they were handled/sampled, if they were sedated, if they were euthanized, and if so, under which procedure?

Lines 81-82 – This sentence is hard to follow as written. Please rephrase. Please also include here which/how many of the original localities you retained in the current study.

Line 94: “Each session consisted in three” should read “Each session consisted of three”

Line 94-95: Was the sampling period the same for P1?

Table 1 – Why do some localities show ? for trapping effort?

Lines 113-117 – Given that the authors have decided to only use localities that were sampled in both P1 and P2 in the analysis, why were new localities sampled in P2? Is it necessary to include those new localities in this MS? If so, that should be clearly justified.

Figure 1 – The maps in the small boxes to the right are very hard to see and interpret, and in general the figure quality is quite poor.

Lines 120-128 – There is not enough detail about statistical methods here. In particular, how was trapping effort accounted for?

Lines 130-138 – I do not see evidence that the Ministry for Public Health of Niger Republic is the equivalent of an Animal Care and Use Committee or Animal Ethics committee. Please include details regarding the capture, handling and release/euthanasia of animals in the methods and/or include the approved protocol as supplementary material. Please note that PLoS Instructions for publications involving animal research REQUIRE details about the IACUC committee used, as well as details regarding anesthesia and euthanasia, to be included.

Line 160-175 – Given that this section refers to Figure 2 repeatedly, I’m confused as to why Figure 2 shows abundance data while this section only refers to presence/absence data. If abundance data is relevant, please discuss in the text or adjust the figure to show the data as it’s presented in the text.

Line 162 – Please specify what “significant trapping efforts” means.

Line 165 – Why does it only “appear” that M. natalensis was found in 9 localities? Either they were found or they were not.

Line 176 – There is not enough information in the Figure 2 caption to be able to interpret the figure. Details about what the pie charts mean and how they were calculated is missing. It is also unclear how trap efforts varied between the P1 and P1 sessions, and if that was accounted for in abundance estimates? Are these per site or per locality?

Figure 2 - Rather than display the abundances of the two trap types differently, I would suggest combining them and showing site-specific relative community abundances in one pie chart per site to highlight the changes between P1 and P2 more clearly. Although the analyses were performed separately due to differences in catch rate, presenting the small mammal composition at a site is not best displayed by separating them in the figure.

Table 3 – Appears to be missing.

Table 4 – This table is incomprehensible as currently displayed. What do the column headings mean? What do the values mean? What are the stats related to and what are they? What is the message of this table?

Line 225 – This is misleading as not only did Mus musculus not “appear to expand massively” between P1 and P2, it did not expand at all if the number of positive sites is the metric used.

Line 236 – Please clarify what is meant by “our observational device”.

Line 237 – ‘Support’ should be used in place of ‘confirm’.

Lines 238-239 – How would an efficient vs. inefficient colonization be distinguished? What characteristics would be observed if Rattus rattus was not showing a ‘quite efficient colonization’ of the city vs. what was observed here?

Line 243-255 – What about features of rodent biology might make this neighborhood make it more suitable to M. natalensis over Rattus rattus?

The discussion about how R. rattus is spreading in the city is too light on alternative explanations other than the type of building/building material. Are there other features of the way the city is connected/used that might explain this? There is extensive literature regarding rodent invasion in other cities that should be referred to here.

Lines 259-260 – The claim that Rattus rattus presence/abundance is negatively associated with M. natalensis should be statistically tested. Please include in the analysis.

Line 272-273 – Seoul virus is properly called Orthohantavirus seoulense (ICTV) and should be italicized. Otherwise suggest referring to only the common name of Seoul virus as Seoul virus, as what is currently written is neither correct taxonomy or accepted common name.

Line 275 – Similarly, Lassa Mammarenavirus should be referred to as Mammarenavirus lassaense (ICTV) or Lassa virus.

Lines 281-283 – Please include references to support this statement.

**Do you want your identity to be public for this peer review?** For information about this choice, including consent withdrawal, please see our Privacy Policy

Reviewer #1: No

Reviewer #2: No

Reviewer #3: No

---

## [Author Response · Author response to Decision Letter 1]

16 Apr 2025

For Reviewer #1

We would like to thank the reviewer for his.her positive and encouraging feedback on our work and manuscript. We are very grateful that his.her found our work well explained and justified. We appreciate his.her constructive suggestions, which we hope have helped us to improve the clarity and quality of our article. All of his.her comments have been considered and taken into account in the revised version of the manuscript.

---

## [Editor Report · Decision Letter 1]

Dear Dr. Ibrahim Danzabarma,

Thank you for submitting your manuscript to PLOS ONE. After careful consideration, we feel that it has merit but does not fully meet PLOS ONE’s publication criteria as it currently stands. Therefore, we invite you to submit a revised version of the manuscript that addresses the points raised during the review process.

We look forward to receiving your revised manuscript.

Kind regards,

Javier delBarco-Trillo, Ph.D.

Academic Editor

PLOS ONE

Journal Requirements:

Additional Editor Comments :

The authors have addressed most of the reviewers’ comments satisfactorily, so I will not send the manuscript for another round of reviews. However, I have identified several issues that still need to be addressed, some of them involving the new text in the revision.

For specific comments referring to the text, the listed lines below refer to the version with Track changes.

Consider if some of the methodological information you offered in responses #19 and #21 shouldn’t be incorporated into the Methods section. You make good points, and such points being stated in the Introduction may minimize similar questions from other readers.

Line 39, as there are two, it should be “genera” and not “genus”. But I would refer to the cosmopolitan species, basically only three (brown and black rat and house mouse), and not to the genera, as between these two genera you have over 100 species and most of them are not cosmopolitan.

Line 81, correct “These sites households”

Line 85, correct “with an effort of sampling 2 to 4 sites per locality”

Line 98: “space”. Same singular/plural inconsistency in line 103: it should be “a new baited trap”

Lines 144-148: “yellow points indicate localities sampled at different sites across periods, and green points show new localities sampled in Period 2” is a bit confusing. Do you mean that “yellow points indicate localities sampled in Period 1 that were not resampled in Period 2”?

Table 2. Check the title row, there is a problem with it.

Line 202, to address reviewer 3’s comment, I would consider changing “significant” for “substantial”, “high”, “strong” or a similar term that denotes the meaning intended while avoiding any confusion with the statistical usage of the term “significant”.

Line 205, something is missing in “both Sh and Wm traps, it appears that M. natalensis”

Lines 246-249, please check the quotations and make sure they are being used appropriately and consistently. I think in line 228 you want to write “ “TN” stand for the…”, as “N” is defined later.

Please correct small typos. For example, in line 275 remove the period after “Table”

In many places, shouldn’t it be “Table S1” instead of “S1 Table”?

Live 284: I agree with Reviewer 3 that “observational device” is a confusing term even if correct, as many readers will primarily associate “device” with a physical object. Please use other term, e.g. “strategy” or “approach”. Or maybe you don’t need “of our observational device” at all?

Line 287, avoid using “quite” to add uncertainty

Line 307, instead of “unknown and still to be investigated in details” you can use “to be investigated”

Line 312, “Fig. S1”

Lines 327 and 330, most readers won’t know what ICTV stand for, and probably will appreciate the common name.

Line 359, “(DOC)”?

Lines 355 and 360, why “Table.” and “Fig:”?

Line 360, again, please go through manuscript and pay attention to small details. Here, do not italicize the “and” between species names

Line 362, “The red dots are for negative ratios and the blue dots for positive ratios”

Line 363: “Spearman”

Fig S1 legend, you use both “figure” and “frame” to refer to the panels. Use “panel” instead

Fig S1, I find the left panel very confusing. You say you want to portray the change between the two periods, but there is no clear indication about the two periods. The reader will think there is something important about the two colors, but they simply identify points over or under 0, but that’s quite confusing. You could have a solid line at 0, which would easily show points above and under. As for the right panel, please use common notation with “r = -.0671, p < 0.01” or whatever the values are.

Table S1, you are providing p values, so what’s the point of adding asterisks or ns? That’s redundant.

References: please check for consistency across references. For example, in many references you include the doi (correctly), whereas in others you add a website (even though a doi is available).

---

## [Author Response · Author response to Decision Letter 2]

9 May 2025

We sincerely thank the Editor for his constructive comments on the revised version of our manuscript. We appreciate very much the time and attention devoted to improving our work.

We are pleased that the revised version of our manuscript has addressed satisfactorily most of the reviewers' comments. We would like also to thank you for the opportunity you give us to submit a further revised version. According to the editor’s comments, we have thoroughly reviewed the entire manuscript and made the necessary changes, including the newly added sections and the list of references as requested. All modifications and corrections made are available in the re-revised version of the manuscript named “Revised Manuscript with Track Changes.docx”.

Please find below both detailed response to each comment and precise indication of the changes made in the revised manuscript, referring to the line numbers in the annotated version.

We hope that these revisions fully address the latest editor’s concerns.

---

## [Editor Report · Decision Letter 2]

Native-to-invasive rodent species turn-over within African cities: the example of Niamey, Niger

PONE-D-24-55809R2

Dear Dr. Ibrahim Danzabarma,

We’re pleased to inform you that your manuscript has been judged scientifically suitable for publication and will be formally accepted for publication once it meets all outstanding technical requirements.

Kind regards,

Javier delBarco-Trillo, Ph.D.

Academic Editor

PLOS ONE

Additional Editor Comments (optional):

When possible, please correct these further points:

Line 79, “included” instead of “including”

Line 101, it should be “morning: any successful”

The Table 3 legend still needs to be corrected. I think the beginning of the second sentence should be “”TN” stands for…”.

---

## [Editor Report · Acceptance letter]

PONE-D-24-55809R2

PLOS ONE

Dear Dr. Ibrahim Danzabarma,

I'm pleased to inform you that your manuscript has been deemed suitable for publication in PLOS ONE. Congratulations! Your manuscript is now being handed over to our production team.

Kind regards,

on behalf of

Dr. Javier delBarco-Trillo

Academic Editor

PLOS ONE